# Antimicrobial stewardship: Attitudes and practices of healthcare providers in selected health facilities in Uganda

Isaac Magulu Kimbowa[1]*, Jaran Eriksen[2,3], Mary Nakafeero[4], Celestino Obua[5], Cecilia Stålsby Lundborg[3], Joan Kalyango[6], Moses Ocan[1]

1 Department of Pharmacology and Therapeutics, Makerere University College of Health Sciences, Kampala, Uganda, 2 Unit of Infectious diseases/Venhälsan, Stockholm South Hospital, Stockholm, Sweden, 3 Department of Global Public Health, Karolinska Institutet, Stockholm, Sweden, 4 School of Public Health, Makerere University College of Health Sciences, Kampala, Uganda, 5 Mbarara University of Science and Technology, Mbarara, Uganda, 6 Department of Pharmacy, Makerere University College of Health Sciences, Kampala, Uganda

* jakemagulu@gmail.com

**Data Availability Statement:** All relevant data are within the paper and its Supporting Information files. They have also been uploaded to https://osf.io/jqbzt/.

## Abstract

Though antimicrobial stewardship (AMS) programmes are the cornerstone of Uganda's national action plan (NAP) on antimicrobial resistance, there is limited evidence on AMS attitude and practices among healthcare providers in health facilities in Uganda. We determined healthcare providers' AMS attitudes, practices, and associated factors in selected health facilities in Uganda. We conducted a cross-sectional study among nurses, clinical officers, pharmacy technicians, medical officers, pharmacists, and medical specialists in 32 selected health facilities in Uganda. Data were collected once from each healthcare provider in the period from October 2019 to February 2020. Data were collected using an interview-administered questionnaire. AMS attitude and practice were analysed using descriptive statistics, where scores of AMS attitude and practices for healthcare providers were classified into high, fair, and low using a modified Blooms categorisation. Associations of AMS attitude and practice scores were determined using ordinal logistic regression. This study reported estimates of AMS attitude and practices, and odds ratios with 95% confidence intervals were reported. We adjusted for clustering at the health facility level using clustered robust standard errors. A total of 582 healthcare providers in 32 healthcare facilities were recruited into the study. More than half of the respondents (58%,340/582) had a high AMS attitude. Being a female (aOR: 0.66, 95% CI: 0.47–0.92, *P < 0.016*), having a bachelor's degree (aOR: 1.81, 95% CI: 1.24–2.63, *P < 0.002*) or master's (aOR: 2.06, 95% CI: 1.13–3.75, *P < 0.018*) were significant predictors of high AMS attitude. Most (46%, 261/582) healthcare providers had fair AMS practices. Healthcare providers in the western region's health facilities were less likely to have a high AMS practice (aOR: 0.52, 95% CI 0.34–0.79, *P < 0.002*). In this study, most healthcare providers in health facilities had a high AMS attitude and fair AMS practice.

**Funding:** This work was supported by Makerere University-Swedish International Development Agency (SIDA) collaboration (Sida PI0010). The funders never participated in the study design, data collection and analysis, decision to publish, or manuscript preparation.

**Competing interests:** The authors have declared that no competing interests exist.

**Abbreviations:** ABR, Antibacterial Resistance; AMS, Antimicrobial Stewardship; ASP, Antimicrobial Stewardship Programmes; WHO, World Health Organisation; GOU, Government of Uganda.

# Introduction

Antimicrobial resistance (AMR) is a global public health threat caused by the misuse of antibacterial agents in human, animal, and environmental sectors [1,2]. Antibacterial misuse involves prescribing antibacterial agents when not needed, while antibacterial overuse involves inappropriate or unnecessary taking of antibacterials [3–5]. Several international, national, and professional organisations, including the World Health Organization (WHO), have called for the establishment of antimicrobial stewardship as a strategy to promote optimal antibacterial use in the human, veterinary, and agricultural sectors in order to reduce the transmission and development of antimicrobial resistance (AMR) [6–8].

Antimicrobial stewardship (AMS) is a set of synchronised interventions that optimise antibacterial use to generate the best clinical outcome, increase patient safety, and reduce the risk of AMR development [9–11]. Together with infection prevention and control (IPC), medicine and patient safety, AMS is one of three "pillars" of an integrated strategy used in strengthening health systems [10]. Adopting AMS interventions in health facilities is critical in supporting healthcare providers with tools and systems in optimising antibacterial use, reducing the transmission and colonisation of multidrug-resistant bacteria, and lowering the incidence of antimicrobial-related adverse events [12]. Additionally, the principles of AMS are extensively applied throughout the One-health approach in optimising antibacterial use in both animal and agriculture sectors, where the emphasis is put on judicious and prudent antibacterial use to avoid the spread and development of antibacterial resistance [10,13]. Furthermore, following the approval of the Global Action Plan (GAP) on AMR, member states of the WHO committed themselves to the development and implementation of National Action Plans (NAPs) on AMR [3,14,15]. Establishing AMS programmes has been prioritised in all national action plans on AMR as a critical objective for optimising antibacterial use [10]. In developing NAPs on AMR, member states of WHO were encouraged to involve relevant stakeholders in different sectors, including institutions, health professionals, policymakers, and patients [3]. Over 117 countries have established NAPs on AMR, with varying stages of implementation of antimicrobial stewardship (AMS) programmes [3,14,15]. According to a recent systematic review, only seven African countries had NAPS on AMR, and among these, only three countries (Kenya, South Africa, and Tanzania) were implementing AMS programmes [11]. However, the same review identified that AMS activities were implemented in countries with neither NAPs nor AMS programmes [11]. On the other hand, previous studies have shown that the adoption of AMS activities is dependent also on the healthcare providers' attitudes and practices, which were affected either by the top-down approach of government policy implementation or bottom-up participation of healthcare providers in policy development [8,16].

Previous studies on AMS showed that healthcare providers' attitudes and practices on AMS varied significantly throughout most countries like Nigeria, Zambia, and Ethiopia [17]. In most health facilities, healthcare providers had a casual and lax attitude towards AMS implementation following its introduction through a top-bottom approach to policy implementation [17]. In addition, despite having good Knowledge about AMS, most healthcare providers did not agree that antibacterials were misused or that AMR was a significant problem in their institutions [17]. Several studies show that the top-down approach has limited the implementation of AMS programmes and activities to only specialised hospitals, thus disregarding community hospitals whose major labour force are allied healthcare professionals [16–18]. As a result, this has limited the application of previous findings on antimicrobial stewardship attitudes and practices to only specialised health care facilities and a particular group of healthcare providers. The exclusion of community health facilities in previous studies in low-and-

middle-income countries (LMICs) has limited the generalizability of their findings on AMS attitudes and practices of healthcare providers in their countries [19].

Several one-health initiatives have trained healthcare providers in health facilities and communities on implementing antimicrobial stewardship in both the human and animal sectors in Uganda [13]. Additionally, when Uganda drafted its National Action Plan (NAP) on AMR for 2018 to 2023, it placed a greater emphasis on a bottom-up approach that included healthcare providers in regional referral centres, general hospitals, and private not-for-profit (PNFP) organisations, as well as other stakeholders to promote AMS [20]. Despite the Ministry of Health's continued engagement with heads of health institutions in strengthening existing medicine and therapeutic committees and antimicrobial stewardship programmes, the attitudes and practices of healthcare providers towards AMS remain unknown in Uganda [13,20]. Therefore, the current study investigated healthcare providers' attitudes and practices towards AMS and associated factors in Uganda's regional referral hospitals, general hospitals, and private-not-for-profit (PNFPs) health facilities.

## Materials and methods

### Study design and setting

We conducted a cross-sectional study from October 2019 to February 2020 among healthcare providers at regional referral hospitals, general hospitals, and PNFPs. Uganda's health system is integrated with about 6,937 health facilities, where 45% are public-owned, 40% are private for-profit (PFP), and 15% are private-not-for-profit (PNFPs) [21,22]. The public health system is hierarchical, referral-based, and provides free health services at all levels of delivery [22,23]. The composition of public health facilities in the country includes; two national referral hospitals, 16 regional referral hospitals, 47 general hospitals, 166 level IV health centres, 962 level III health centres, and 1321 level II health centres and 1558 clinics [21,23,24]. The two National referral hospitals are urban teaching hospitals with a bed capacity ranging from 600 to 1500. Regional referrals hospitals are teaching hospitals located in urban centres with a bed capacity ranging from 250 to 600 beds, while general hospitals are community-based, all have 100 beds. There are 1009 health facilities, four tertiary hospitals, followed by 40 general hospitals that serve community settings and 955 health centres [22,24]. Due to government funding, they offer cheaper, subsidised services than PFP [21,23]. There are 2976 health facilities in the PFP healthcare systems of Uganda [21].

The study was conducted at selected regional referral, general hospitals and tertiary PNFP hospitals in all regions of Uganda. The hospitals above were selected because their healthcare providers and those in the health centres and communities had received training on antimicrobial stewardship through several one-health initiatives [13]. Additionally, the government is strengthening health systems by operationalising medicines and therapeutic committees to strengthen supply chain management, antimicrobial stewardship programmes, and pharmacovigilance [20,25].

### Study population

The study population included hospital directors, nursing officers, heads of department (Medicine, Paediatrics, Surgery and Obstetrics and Gynaecology), medical officers, clinical officers, pharmacy technicians, pharmacists, and medical specialists. In this study, pharmacy technicians, referred to as dispensers had a pharmacy diploma while pharmacists had a bachelor's degree. Clinical officers, referred to community health officers, had a diploma in clinical medicine, while medical officers and medical specialists had a bachelor's degree and a master's degree in medicine, respectively. The study included only permanent staff who had worked for

more than two years. Part-time staff, medical residents, interns and those who had worked for only one year after transferring to the health facility were not included.

## Sample size determination

The required sample size was determined using a single population proportion formulae [26]. We took the proportion to be 50% (P = 0.5) to have a maximum sample size possible with the formulae since there were no previous studies on AMS attitudes and practices. Using a population of 42500 healthcare providers with a 5% margin of error, we obtained a sample size of 381. We adjusted for clustering by multiplying the sample size (381) obtained by 1.5 and adjusted for a non-response rate of 35.4%, thus generating a sample size of 768 health providers. The study targeted eight regional referrals, 32 general hospitals and three tertiary PNFP to achieve the required sample size. In each facility, we targeted participation of 17 to 24 health providers to reach the estimated sample size.

## Sampling procedure

The sampling frame consisted of 16 regional referrals, 47 general hospitals and four tertiary PNFPs. The health facilities were selected because their healthcare providers had previously received AMS training from numerous One-health initiatives.

We selected eight regional referral facilities out of 16 (50%) using simple random sampling (lottery method). We then selected 32 general hospitals out of 47 (68.1%) using simple random sampling (lottery technique). When the selection procedure was completed, four general hospitals were selected for each of the eight regional referral hospitals in each of the country"s regions. Lastly, we randomly selected three out of four (75%) tertiary PNFPs hospitals. We sought administrative clearance (consent to participate) of all the selected health facilities (eight regional referrals, 32 general hospitals and three tertiary PNFPs). The study received administrative clearance from eight selected regional referral hospitals, only 21 out of 32 general hospitals, and three tertiary PNFPs.

We selected healthcare providers using a proportionate number to size. We targeted 224 nurses, 192 clinical officers, 32 pharmacy technicians, 194 medical officers, 32 pharmacists, 64 medical experts, and 32 laboratory technicians out of the needed 768 healthcare providers. We computed the number of different professionals to be selected from each facility by dividing the number of people in a specific profession by the total number of health professionals to obtain the fraction of that profession at the facility. This fraction was then multiplied by the total number of health professionals to be sampled from the health facility. The different numbers of healthcare providers per health facility selected in the study are shown (S1 Fig supplementary information).

## Survey tool development

We conducted an extensive literature review with keywords related to antimicrobial stewardship, attitude and practices to generate a pool of questions. The questionnaire items on AMS attitude included healthcare providers' familiarity with AMS, the effectiveness of AMS in improving patient outcomes, patient safety, and reducing the spread of antibacterial resistance [17,27]. AMS practice items included; adherence to standard treatment guidelines, culture and susceptibility testing, avoidance of excessive use of broad-spectrum antibacterials, and surgical antibacterial prophylaxis [17,28–31]. The pharmacologist prepared the initial version of the instrument in English since it targeted only healthcare practitioners whose formal language of practice is English. We invited specialists in public health (1), epidemiology (1), microbiology (1), pharmacy (1), and pharmacology (1) to review and modify the instrument to improve the

clarity of each item's questions, ease of comprehension, and style and structure of the questionnaire.

Furthermore, specialists agreed on the AMS attitudes and practices questions' readability, clarity, and comprehensiveness. We pilot-tested the questionnaire in four hospitals with 20 healthcare practitioners (doctors (6), nurses (4), allied health workers (8), and pharmacists (2). The respondents gave feedback on questions that needed reformulation, rewording, as well as those that were difficult to understand. We tested the reliability of the piloted tool by conducting an alpha Cronbach's coefficient, where AMS attitude was 0.9268 and 0.762 for AMS practice. After approval from the experts, the final version of the questionnaire contained arranged attitude and practice questions according to the respondents' breadth and depth of understanding of their particular hierarchies (Bloom, 1956) (S1 Appendix supplementary material).

## Variables

The outcome variables were attitude and practices on AMS. AMS attitude and practice scores were generated as the sum of points in each of the 12 questions. The questions on AMS attitude were Likert type, and responses scores were coded from 1 to 5 (strongly disagree = 1, disagree = 2, neither agree nor disagree = 3, agree = 4, strongly agree = 5), giving a possible minimum of 12 and a possible maximum of 60 points in all 12 questions. The questions on AMS practices required "yes" (coded as 1) or "no" (coded as 0) responses and thus had a minimum possible score of 0 and a maximum possible score of 12. The study graded healthcare providers' AMS attitude and practice scores using modified Bloom's categorisation [32]. According to this study, AMS attitude or practice scores had a "high" score if they ranged between 80 and 100% (47–60) points for attitude and over ten points for practice, "fair " if the score was between 50 and 79% (30–46 points for AMS attitude, and 6 to 9 points for AMS practices), and "low" if the score was less than 50% (30 points for attitude and less than 6 points for AMS practice). Independent variables included both social-demographic (sex, age, years of experience, level of academic training, healthcare provider's profession) and hospital characteristics such as the type of health facility (general, regional referral, and private-not-for-profit), nature of health facility (teaching and non-teaching hospitals) and bed capacity.

## Data collection

Data were collected using an interviewer-administered questionnaire. Before data collection, research assistants, comprising medical officers, pharmacists, nurses, and hospital biostatisticians, were trained on the questionnaire. The survey questionnaire had sections on: (i) hospital characteristics; (ii) respondents' socio-demographic characteristics; (iii) AMS attitude; (iv) AMS practice.

To recruit study respondents, research assistants used phone calls, text SMS, emails and letters to invite all selected healthcare providers from the Departments of Medicine, Paediatrics, Surgery, Obstetrics and Gynaecology, Private, Outpatients, Pharmacy, and Laboratory for the interview. We used a list of health workers obtained from the medical director's office or heads of departments. Research assistants made reminder phone calls to the potential study participants to increase participation. We informed every recruited respondent that their participation was voluntary. After accepting to participate, the research assistant provided a brief introduction of the study, objective, and procedures and informed the respondent of anonymity, confidentiality and all declarations. Responding to the questionnaire took 25 to 30 minutes. The respondents were compensated for their time and transport. We collected data in each health facility for two weeks from November 2019 and February 2020.

## Data processing and management

The research assistant evaluated every questionnaire for accuracy and completeness at the end of each day's data collection. During fieldwork and data cleaning, we performed a thorough case analysis to detect missing data on variables in the questionnaires. We dropped any questionnaires containing significant missing data on study variables during the data cleaning process. We utilised EpiDATA manager to conduct double data entry and validation during which data collection tools were entered twice by different data entrants, which we reconciled to detect any differences or discrepancies. Those variables which diverged from each other were thoroughly checked against the original questionnaires and harmonised accordingly. We performed the data validation during entry until the original and entered files were similar to each other.

## Data analysis

All data collected was analysed using STATA 15.1 (Stata Corp, Texas, USA). The study summarised categorical variables using proportions, and it further described continuous variables using means and standard deviations or medians and interquartile ranges (IQR). Socio-demographic variables associated with the AMS attitude and practice scores were determined using ordinal logistic regression in bivariable and multivariable analysis. After testing the association between AMS attitudes or practices with social demographics, two variables (nature of teaching health facility and bed capacity) had a variance inflation factor (VIF) greater than 10, which indicated the presence of multicollinearity. The study chose the type of teaching health institution over bed capacity because it had a lower Bayesian information criterion (BIC) value.

We included all variables with a p-value less than 0.2 at bivariable analysis in the multivariable analysis. Two independent variables, the region of Uganda and the health facility department, violated the proportional odds assumption. There was no significant difference between non-proportional and proportional odds models using BIC. So all ordinal logistic regression used proportional odds models. We included age and sex as universal confounders even when they did not reach the 0.2 significance criterion in the multivariable model using the backward selection technique, along with all variables with a p-value less than 0.2 in the bivariable model. The dependent variable had three categories; "low," "fair," and "high" attitude or practice scores of AMS. Independent variables were assessed for statistical interactions and confounding. In this study, variables with p-values less than 0.05 were considered statistically significant in the final model, where sex and age were used as universal confounders. Associations of AMS attitudes and practices were presented using odds ratios and their corresponding 95% confidence intervals. The research used clustered robust standard errors to account for health facility clustering.

## Ethical considerations

The protocol received ethical clearance from the Makerere University School of Biomedical Sciences Higher Degree Research and Ethics Committee (reference number SBH-HDREC-624). The study got further ethical approval from the Uganda National Council of Science and Technology (UNCST) and gave ethical approval (reference number HS339ES). Heads of health facilities granted the study protocol administrative clearance, permitting the principal investigator to conduct the study among healthcare providers in all participating health facilities. Before responding to the questionnaire, written informed consent was obtained from all targeted respondents. We kept all the questionnaires collected from the survey in lockable lockers

for confidentiality. All information about the healthcare providers was de-identified to ensure anonymity.

## Results

### Socio-demographic characteristics of study respondents

Of the 768 potential study respondents contacted for enrolment from 32 health facilities, 582 completed the study questionnaire (76%, 582/768). More than half of the study respondents were females (57%, 333/582). The overall median age of the respondents was 38 (IQR, 34–43) years. Most of the respondents (42%, 246/582) were between 30 and 39 years old. Over half (56%, 327/582) of the healthcare providers had a diploma level of academic training. Most healthcare providers (44%, 258/582) had worked for more than ten years (Table 1).

### Antimicrobial stewardship attitudes of healthcare providers in health facilities in Uganda

More than half of the healthcare providers enrolled in this study had a high AMS attitude (58%, 340/582). Pharmacists had the highest mean AMS attitude scores compared to all healthcare providers. Nurses had the least mean AMS attitude scores. Healthcare providers agreed that implementation of AMS strategies in health facilities minimises the risk of antibacterial resistance development (87%, 507/582), decrease patient length of stay (85%, 496/582),

**Table 1. Characteristics of study respondents (N = 582).**

| Description | Frequency (N = 582) | Percentage (%) |
|---|---|---|
| **Sex** | | |
| Females | 333 | 57.2 |
| Males | 249 | 42.8 |
| **Age (years)** | | |
| 20–29 | 96 | 16.5 |
| 30–39 | 246 | 42.3 |
| 40–49 | 171 | 29.4 |
| 50+ | 69 | 11.9 |
| **Level of academic training** | | |
| Diploma | 327 | 56.2 |
| Degree | 191 | 32.8 |
| Masters | 64 | 11 |
| **Years of experience** | | |
| Less than five years | 184 | 31.6 |
| 5 < 9 | 140 | 24.1 |
| 10+ | 258 | 44.3 |
| **Healthcare providers** | | |
| Nurses | 199 | 34.2 |
| Pharmacy Technicians (PTs) | 30 | 5.2 |
| Clinical Officers (COs) | 136 | 23.4 |
| Medical Officers (MOs) | 121 | 20.8 |
| Pharmacists (P) | 24 | 4.1 |
| Medical specialist (MS) | 50 | 8.6 |
| Laboratory technicians (LTs) | 22 | 3.8 |

**Table 2. Attitudes of healthcare providers on antimicrobial stewardship (AMS) in health facilities in Uganda (N = 582).**

| | Healthcare providers in selected health facilities (N = 582) | | | | | | | | |
|---|---|---|---|---|---|---|---|---|---|
| | Nurses | PT | CO | MO | P | MS | LT | Total | |
| | (n = 199) | (n = 30) | (n = 136) | (n = 121) | (n = 24) | (n = 24) | (n = 22) | 580 | |
| **Antimicrobial stewardship (AMS) attitudes** | (%) | (%) | (%) | (%) | (%) | (%) | (%) | (100) | P-Value |
| I know what AMS means | 111 (55.8) | 20 (66.7) | 78 (57.3) | 84 (69.4) | 24 (100.0) | 33 (66.0) | 14 (63.7) | 364 (62.5) | 0.001 |
| I am familiar with AMS goals | 74 (37.2) | 13 (43.3) | 54 (39.7) | 55 (45.5) | 16 (66.6) | 29 (58.0) | 7 (31.8) | 248 (42.6) | 0.001 |
| AMS is essential in this health facility | 153 (76.9) | 26 (86.7) | 104 (76.5) | 102 (84.3) | 22 (91.7) | 41 (82.0) | 13 (59.1) | 461 (79.2) | 0.004 |
| AMS involves appropriate selection of antibacterials | 137 (68.8) | 23 (76.7) | 104 (76.4) | 97 (80.2) | 22 (91.6) | 37 (74.0) | 12 (54.6) | 432 (74.0) | 0.001 |
| AMS involves optimal antibacterial administration | 144 (72.4) | 22 (73.3) | 101 (74.3) | 93 (76.9) | 22 (91.6) | 37 (74.0) | 15 (68.2) | 434 (75.0) | 0.023 |
| AMS interventions can improve patient outcomes | 170 (85.4) | 28 (93.3) | 114 (83.9) | 104 (86.0) | 23 (95.8) | 44 (88.0) | 19 (86.3) | 502 (86.0) | 0.094 |
| AMS strategies can reduce the problem of antimicrobial resistance | 172 (86.4) | 28 (93.4) | 117 (86.0) | 105 (86.7) | 24 (100.0) | 42 (84.0) | 19 (86.4) | 507 (87.1) | 0.117 |
| AMS can reduce the length of hospital stay | 168 (84.5) | 28 (93.3) | 112 (82.4) | 102 (84.3) | 24 (100.0) | 44 (88.0) | 18 (81.8) | 496 (85.2) | 0.417 |
| AMS practices can increase appropriate antibacterial use | 147 (73.8) | 27 (90.0) | 114 (83.8) | 100 (82.6) | 24 (100) | 44 (88.0) | 18 (81.9) | 474 (81.4) | 0.001 |
| AMS strategies can decrease the incidence of *Clostridium difficile* rates | 139 (69.9) | 20 (66.7) | 87 (63.9) | 89 (73.5) | 22 (91.7) | 32 (64.0) | 16 (72.8) | 405 (69.6) | 0.289 |
| Source of information on AMS practices. | 61 (30.7) | 6 (20.0) | 38 (27.9) | 30 (24.8) | 5 (20.8) | 13 (26.0) | 5 (27.1) | 158 (27.1) | 0.943 |
| Additional staff education on AMS is needed | 171 (85.9) | 28 (93.3) | 116 (85.3) | 103 (85.1) | 24 (100) | 45 (90.0) | 18 (81.8) | 505 | 0.057 |
| AMS attitude scores reported as mean and standard deviation (SD) in each profession group | 44.5±11.4 | 47.8±9.2 | 45.3±11 | 47.2 ±10.9 | 52.7±4 | 47.5±11 | 43.9 ±12.6 | 46±11 | |

[a]PT: Pharmacy technician, CO: Clinical officer, MO: Medical officer, P: Pharmacist, MS: Medical Specialists LT: Laboratory technician.

[b]A Likert scale rated from one (strongly disagree) to 5 (strongly agree) and *show significant difference at P < 0.05.

improve patient outcomes (82%, 492/582) and increase appropriate antibacterial use (81%, 474/582) (Table 2).

## Factors associated with antimicrobial stewardship attitude among healthcare providers in health facilities in Uganda

In a bivariable analysis, AMS attitude amongst healthcare providers was significantly associated with level of academic training (P = 0.002), hospital department (P = 0.006), sex (P = 0.005), and geographical region (P = 0.001). After controlling for education and region, the multivariable logistic regression model showed that females (AOR: 0.66, 95% Cl: 0.47–0.92) were significantly less likely to have high AMS attitude scores than males after controlling for education and the region. Healthcare providers with a bachelor's degree (AOR: 1.81, 95% Cl: 1.24–2.63) were 1.8 times significantly more likely to have high AMS scores than those with diplomas. Similarly, healthcare providers with a master's degree (AOR: 2.06: 95% Cl: 1.24–2.63) were 2.1 times significantly more likely to have high AMS attitude scores than those with diplomas (Table 3).

**Table 3. Predictors of antimicrobial stewardship (AMS) attitudes amongst healthcare providers in health facilities in Uganda (N = 582).**

|  | Low score | Fair score | High scores | COR | AOR | P-value |
|---|---|---|---|---|---|---|
|  | (n = 52) | (n = 190) | (n = 340) | (95% CI) | (95% CI) |  |
|  | n(%) | n(%) | n(%) |  |  |  |
| **Age (years)** |  |  |  |  |  |  |
| 20–29 | 13(25) | 29(15.3) | 54(15.9) | 1 |  |  |
| 30–39 | 16(30.8) | 77(40.5) | 153(45) | 1.48 (0.91–2.39) | 1.46 (0.90–2.37) | 0.121 |
| 40–49 | 18(34.6) | 62(32.6) | 91(26.8) | 1.06 (0.64–1.77) | 1.05(0.63–1.74) | 0.862 |
| 50+ | 5(9.6) | 22(11.6) | 42(12.4) | 1.59 (0.83–3.02) | 1.57(0.83–2.99) | 0.168 |
| **Sex** |  |  |  |  |  |  |
| Male | 23(44.2) | 97(51.1) | 213(62.6) | 1 | 1 |  |
| Female | 29(55.8) | 93(48.9) | 127(37.4) | 0.65 (0.46–0.91) | 0.66 (0.47–0.92) | *0.016 |
| **Level of academic training** |  |  |  |  |  |  |
| Diploma | 36(69.2) | 124(65.3) | 167(49.1) | 1 | 1 |  |
| Degree | 13(25) | 51(26.8) | 127(37.4) | 1.77 (1.21–2.58) | 1.81 (1.24–2.63) | *0.002 |
| Masters and above | 3(5.8) | 15(7.9) | 46(13.5) | 1.96 (1.05–3.65) | 2.06 (1.13–3.75) | *0.018 |

COR: Crude Odds Ratio, AOR: Adjusted Odds Ratio, CI: Confidence Interval.

*show significant difference at $p < 0.05$.

## AMS practices among healthcare providers in health facilities in Uganda

Most respondents (47%, 261/582) had a fair AMS practice score in this study. Medical officers had the highest mean AMS practices, while pharmacists had the least mean AMS practice scores.

The most-reported AMS practices implemented in health facilities included; documenting antibacterial use (90%, 501/560); using standard treatment guidelines to initiate effective antibacterial treatment (78%, 438/560); and complying with culture and susceptibility results (76%, 425/560) (Table 4).

## Factors associated with antimicrobial stewardship practices among healthcare providers in health facilities in Uganda

In bivariate analysis, AMS practice scores were significantly associated only with the region (P = 0.003). In the multivariable model, after having adjusted for confounders, AMS practice scores of healthcare providers in the western region (AOR: 0.52, 95% CI: 0.34–0.79) were significantly lower than those in the central region (Table 5).

## Discussion

Healthcare providers are more likely to their attitudes and practices if they are involved in the policymaking process and agree with the proposed changes [33]. The Ministry of Health used a bottom-up strategy, where its actively engaged healthcare providers in operationalising the implementation of AMS programmes and strengthening Medicines and Therapeutics committees in health facilities [20]. Despite the continued engagement with health facilities, healthcare providers' attitudes and practices and associated factors towards AMS have remained unknown in all four Ugandan regions. This study uses data from an interviewer-administered questionnaire among 582 healthcare providers in 32 health facilities from October 2019 to February 2020 to explore attitudes and practices concerning antimicrobial stewardship (AMS) in selected health facilities in Uganda.

**Table 4. Practices of healthcare providers on antimicrobial stewardship in health facilities in Uganda.**

| | Healthcare providers in selected health facilities (N = 582) | | | | | | | | P-Value |
|---|---|---|---|---|---|---|---|---|---|
| | **Nurse** | **PT** | **CO** | **MO** | **P** | **MS** | **LT** | **Total** | |
| | **(n = 199)** | **(n = 30)** | **(n = 136)** | **(n = 121)** | **(n = 24)** | **(n = 24)** | **(n = 22)** | **580** | |
| **Antimicrobial stewardship (AMS) Practices** | (%) | (%) | (%) | (%) | (%) | (%) | (%) | (%) | |
| Use of standard treatment guidelines | 144 (72.4) | 23 (76.7) | 114 (83.8) | 104 (86.0) | 16 (66.7) | 37 (74.0) | 18 (81.8) | 456 (78.4) | *0.037 |
| Avoid unnecessary broad spectrum antibacterial use | 132 (66.3) | 14 (46.7) | 85 (62.5) | 81 (66.9) | 5 (20.8) | 29 (58.0) | 8 (36.4) | 354 (60.8) | *<0.001 |
| Documenting antibacterial use in patient care | 181 (91.0) | 27 (90.0) | 118 (86.8) | 114 (94.2) | 17 (70.8) | 44 (88.0) | 20 (90.9) | 521 (89.5) | *0.036 |
| Pre-surgical single-dose antibacterial administration | 94 (47.2) | 14 (46.7) | 58 (42.6) | 56 (46.3) | 9 (37.5) | 18 (36.0) | 5 (22.7) | 254 (43.6) | 0.319 |
| Complying with culture and susceptibility results | 150 (75.4) | 21 (70.0) | 98 (72.1) | 93 (76.9) | 18 (75.0) | 45 (90.0) | 9 (40.9) | 434 (74.6) | *0.002 |
| Antimicrobial prescription audit and review | 146 (73.4) | 23 (76.7) | 106 (77.9) | 94 (77.7) | 17 (70.8) | 34 (68) | 12 (54.5) | 432 (74.2) | 0.275 |
| Antibacterial time-out | 141 (70.9) | 19 (63.3) | 94 (69.1) | 85 (70.2) | 16 (66.7) | 34 (68.0) | 9 (40.9) | 398 (68.4) | 0.182 |
| Patient education on antibacterial use | 134 (67.3) | 21 (70.0) | 101 (74.3) | 89 (73.6) | 18 (75.0) | 32 (64.0) | 14 (63.6) | 409 (70.3) | 0.657 |
| Existence of antibacterial use best practices | 107 (53.8) | 14 (46.7) | 89 (65.4) | 78 (64.5) | 14 (58.3) | 25 (50.0) | 10 (45.5) | 337 (57.9) | 0.09 |
| Assessment of antibacterial use (quality and quantity) | 86 (43.2) | 13 (43.3) | 55 (40.4) | 44 (36.4) | 10 (41.7) | 10 (20.0) | 7 (31.8) | 225 (38.7) | 0.111 |
| Measurement of antibacterial use outcomes | 111 (55.8) | 16 (53.3) | 94 (69.1) | 75 (62.0) | 12 (50.0) | 25 (50.0) | 14 (63.6) | 347 (59.6) | 0.121 |
| Use of hospital antibacterial audit data | 88 (44.2) | 8 (26.7) | 58 (42.6) | 54 (44.6) | 6 (25.0) | 23 (46.0) | 8 (36.4) | 245 (42.1) | 0.31 |
| AMS practice scores reported as means and standard deviation (SD)) in each of the professional groups | 7.6 ± 3 | 7.1±2.8 | 7.9±2.9 | 8.0 ± 2.6 | 6.6±3.0 | 7.1±2.7 | 6.1± 3.3 | 7.6 ± 2.9 | |

PT: Pharmacy technician, CO: Clinical officer, MO: Medical officer, P: Pharmacist, MS: Medical specialist, LT: Laboratory technician, SD: Standard deviation.

*shows a significant difference at $p < 0.05$.

More than half (58%) of the respondents had high AMS attitude scores in our study. This finding contrasts with a previous study on AMS attitude conducted in Ethiopia, where 16% of healthcare providers had a high AMS attitude [17]. The variations in AMS attitudes between the two studies could be because of differences in participating health facilities as well as the bottom-up approach of the Ministry of Health involving health facility leaders in the strengthening or operationalisation of AMS programmes and medicine and therapeutics committees in public health facilities and PNFPs. As shown in our study, the bottom-up strategy improved commitment of healthcare providers leading to adoption of interventions that may induce behaviour change. However, a previous study in Ethiopia showed that low AMS could have arisen due to implementing restrictive AMS strategies, which potentially affected the attitude of healthcare providers [34]. While the Ministry of Health in Uganda is in the implementation stages of the NAP on AMR, which places AMS as a critical priority, the high AMS attitude in our study suggests that health facilities could have adopted AMS programmes with strategies that improve healthcare providers' attitudes prior to the Ministry of Health policy intervention.

AMS education covers many subjects, including proper antimicrobial selection and prescription, optimising dosages and duration, and minimising toxicity and side effects to

**Table 5. Predictors of antimicrobial stewardship practices amongst healthcare providers in health facilities in Uganda (N = 582).**

| | Low score | Fair score | High scores | COR | AOR (95% CI) | P-value |
|---|---|---|---|---|---|---|
| | (n = 133) | (n = 261) | (n = 166) | (95% CI) | | |
| | n(%) | n(%) | n(%) | | | |
| **Age** | | | | | | |
| 30–39 | 20(21.7) | 46 (50) | 26(28.3) | 1 | 1 | |
| 20–29 | 52(22.4) | 115(49.6) | 65(28) | 0.99 (0.62–1.6) | 1.02 (0.66–1.61) | 0.901 |
| 40–49 | 45(26.9) | 69(41.3) | 53(31.7) | 0.94 (0.64–1.38) | 0.95 (0.65–1.39) | 0.785 |
| 50+ | 16(23.3) | 31(44.9) | 22(31.9) | 1.11 (0.67–1.87) | 1.15 (0.69–1.91) | 0.593 |
| **Sex** | | | | | | |
| Male | 83(26.1) | 143(45) | 92(28.9) | 1 | 1 | |
| Female | 50(20.7) | 118(48.8) | 74(30.6) | 1.13 (0.75–1.69) | 1.18(0.85–1.62) | 0.313 |
| **Region of Uganda** | | | | | | |
| Central | 37(20.2) | 91(49.7) | 55(30.1) | 1 | 1 | |
| North | 10(15.6) | 29(45.3) | 25(39.1) | 1.52 (0.89–2.60) | 1.48(0.86–2.53) | 0.153 |
| East | 35(20.3) | 80(46.5) | 57(33.1) | 1.13(0.76–1.68) | 1.1 (0.75–1.64) | 0.615 |
| West | 51(36.2) | 61(43.3) | 29(20.6) | 0.54(0.35–0.82) | 0.52 (0.34–0.79) | *0.002 |

COR: Crude Odds Ratio, AOR: Adjusted Odds Ratio, CI: Confidence Interval.

*shows a significant difference at $p < 0.05$.

improve clinical, economic, and microbiological results[35]. As a result, prior research has emphasised the need of employing a multidisciplinary team of highly qualified pharmacists and infectious disease specialists to lead AMS programmes [36]. Our study agrees with previous findings where those with high academic training, like having a bachelor's or master's degree, had significantly higher AMS scores than those with diploma training. After controlling for education, females had lower AMS attitude scores than males. This demonstrates that females (56%) may have comprised a significant proportion of diploma holders who could not obtain AMS training before and during practice experience. The low AMS attitude scores of diploma holders and females could affect the implementation of AMS programmes in terms of comprehensiveness, quality, and adoption to lower community health facilities [37]. These findings suggest the need for curricula on AMS service training for all diploma holders to harmonise their attitude with those of higher qualification regarding AMS to strengthen the multidisciplinary healthcare provider capacity to perform strategies of AMS.

In our study, a third of the respondents reported high AMS practices scores. Our study finding contrasts that of a previous study in Ethiopia, which found that over 70% of healthcare providers have high AMS practices [17]. Despite the high AMS attitude reported in this study, most healthcare providers reported fair AMS practices. However, medical officers had a high mean AMS practice score compared to other healthcare providers. This finding could be an indicator of challenges in implementing the AMS programmes. The absence of national AMS guidelines for health facilities, non-functional microbiology laboratories, and many low-level healthcare cadres employed in health facilities may contribute to this fair AMS practice reported in this study [38].

In this study, reported AMS practices were significantly associated with the geographic region of Uganda. Healthcare providers in the western part of the country were less likely to report high AMS practices than other regions. A previous study conducted in the Western region of Uganda reported a lack of AMS programmes and the need to strengthen infection control practices in Western Uganda's health facilities [39]. This lack of AMS programmes in

health facilities in Western Uganda may explain our study's observed finding. There has been a significant improvement in healthcare infrastructure in the country, where the government has constructed, renovated, and upgraded many health facilities. In addition, more healthcare personnel have been recruited, hence improving the staffing levels to over 70% in most health facilities [40]. However, there is a need for national guidelines on AMS programmes and specific AMS training for healthcare providers in health facilities.

Our finding on the high AMS attitude of pharmacists agrees with previous studies, which have demonstrated that pharmacists have a high positive attitude towards AMS [29,41,42]. However, our study found that pharmacists had a low AMS practice [29]. This finding is similar to a previous study conducted in Zambia, where community pharmacists had low AMS practices concerning AMS [29]. Much as community pharmacists in the Zambian study had a challenge of dispensing antibacterials without prescriptions, lack of providing complete counselling information to patients, pharmacists in our study were less likely to use standard treatment guidelines, avoid unnecessary use of broad-spectrum antibacterials and measure the quality and quantity of antibacterial use in their health facilities. The high AMS attitude of pharmacists in our study could be from previous training on AMS though the low AMS practices could be arising from the limited mandate pharmacist could have as professionals in patient care decision making. Unlike previous studies showing the changing role of a pharmacist inpatient care under antimicrobials stewardship, in Uganda, they are still confined to their traditional function of providing advice on proper antimicrobial utilisation and creating awareness campaigns targeting other healthcare providers about the appropriateness of antimicrobial prescribing and the use of standard treatment guidelines [43,44]. There is a need for policy intervention through the Ministry of Health to strengthen AMS programmes to expand the pharmacist's role under a multidisciplinary team, as reported by several studies.

The study's limitations could be attributed to social desirability, which could have arisen from respondent's responses to different interviewers. We minimised this measurement bias by piloting the questionnaire to minimise ambiguity in questions, rephrasing and rewording the questions. Using an interviewer-administered questionnaire minimised the social desirability effect. However, interviewer bias in this study was minimised by using data collectors/interviewers from the same hospital unit. Part-time and intern healthcare practitioners could not be included in the study since they were not on the permanent employees' lists, even though they prescribed antibacterials. We could have missed responses from this group of persons. The study had a non-response of about 24% of the sampled health providers and a lack of administrative clearance from some facilities, which could have created selection bias. The questionnaire assessed AMS practice using "yes' or 'no' responses, which may have over or underestimated AMS practice among healthcare providers. Our research could not determine the cause and effect relationship of whether high AMS attitude scores also contributed to fair AMS practices. There is a possibility that our findings or conclusions are not generalisable to non-Ugandan or non-East African situations. The study used tools that had been pilot-tested whose reliability and validity was known before data collection, and this potentially reduced the likelihood of under or overestimating AMS practices. Furthermore, the high Cronbach alpha (0.7) indicated the test items' reliability and internal consistency in the tool. The inclusion of healthcare providers of various levels of training and profession from all the regions of Uganda increased the representativeness of this study's findings.

## Conclusion

In this study, most healthcare providers reported a high AMS attitude and fair AMS practices scores. The Ministry of Health should support and regularly monitor the countrywide

implementation of AMS programmes by educating all public hospital healthcare providers, since our study found a significant association between AMS attitude and practices with education levels and geographic location. There is a need for more studies to assess whether adopted AMS programmes exist in these health facilities and the characteristics and challenges of implementing AMS strategies on optimising antibacterial use.

## Supporting information

**S1 Checklist. STROBE statement checklist of items included in reports of cross-sectional studies.**
(DOCX)

**S1 Fig. Flow diagram of the selected health facilities that participated in this study.**
(TIF)

**S1 Appendix. Questionnaire for sub-study II on antimicrobial stewardship attitudes and practices of healthcare providers in selected health facilities in Uganda.**
(DOCX)

**S2 Appendix. Supporting data for the manuscript.**
(DOCX)

## Acknowledgments

The authors thanks Prof Jasper Ogwal Okeng, Dr Jackson Mukonzo, and all Department of Pharmacology and Therapeutics Makerere University staff. The authors want to thank all the healthcare providers who agreed to participate in this study. They also thank all research assistants who collected data from eight regional referrals, 21 general hospitals, and three private-not for-profit facilities. We are grateful to all the 32 healthcare facilities that participated in this study. The authors also like to thank Dr Fred Kitutu and Dr Ronald Kiguba from the Makerere University Department of Pharmacy and Department of Pharmacology and Therapeutics for their survey implementation guidance.

## Author Contributions

**Conceptualization:** Isaac Magulu Kimbowa, Jaran Eriksen, Celestino Obua, Cecilia Stålsby Lundborg, Joan Kalyango, Moses Ocan.

**Data curation:** Isaac Magulu Kimbowa, Jaran Eriksen, Mary Nakafeero, Celestino Obua, Cecilia Stålsby Lundborg, Joan Kalyango, Moses Ocan.

**Formal analysis:** Isaac Magulu Kimbowa, Jaran Eriksen, Mary Nakafeero, Joan Kalyango, Moses Ocan.

**Funding acquisition:** Celestino Obua.

**Investigation:** Isaac Magulu Kimbowa, Cecilia Stålsby Lundborg, Joan Kalyango, Moses Ocan.

**Methodology:** Isaac Magulu Kimbowa, Jaran Eriksen, Mary Nakafeero, Cecilia Stålsby Lundborg.

**Project administration:** Jaran Eriksen, Celestino Obua, Cecilia Stålsby Lundborg, Joan Kalyango, Moses Ocan.

**Resources:** Cecilia Stålsby Lundborg, Joan Kalyango.

**Supervision:** Celestino Obua, Cecilia Stålsby Lundborg, Joan Kalyango, Moses Ocan.

**Validation:** Isaac Magulu Kimbowa, Mary Nakafeero, Joan Kalyango.

**Visualization:** Isaac Magulu Kimbowa, Cecilia Stålsby Lundborg, Joan Kalyango.

**Writing – original draft:** Isaac Magulu Kimbowa.

**Writing – review & editing:** Isaac Magulu Kimbowa, Jaran Eriksen, Mary Nakafeero, Celestino Obua, Cecilia Stålsby Lundborg, Joan Kalyango, Moses Ocan.

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
