## [Decision Letter · Decision Letter 0]

23 Jul 2021

PONE-D-21-10939

Antimicrobial stewardship: Attitudes and practices of healthcare providers in selected health facilities in Uganda

PLOS ONE

Dear Dr. Kimbowa,

Thank you for submitting your manuscript to PLOS ONE. After careful consideration, we feel that it has merit but does not fully meet PLOS ONE’s publication criteria as it currently stands. Therefore, we invite you to submit a revised version of the manuscript that addresses the points raised during the review process.

We look forward to receiving your revised manuscript.

Kind regards,

Elena Ambrosino

Academic Editor

PLOS ONE

Journal Requirements:

2. Please include additional information regarding the survey or questionnaire used in the study and ensure that you have provided sufficient details that others could replicate the analyses. For instance, if you developed a questionnaire as part of this study and it is not under a copyright more restrictive than CC-BY, please include a copy, in both the original language and English, as Supporting Information. Moreover, please include more details on how the questionnaire was pre-tested, and whether it was validated.

5. We note you have included a table to which you do not refer in the text of your manuscript. Please ensure that you refer to Table 5 in your text; if accepted, production will need this reference to link the reader to the Table.

6. Please upload a copy of Supporting Information S5 which you refer to in your text on page 292.

Reviewers' comments:

Reviewer's Responses to Questions

**Comments to the Author**

1. Is the manuscript technically sound, and do the data support the conclusions?

Reviewer #1: Partly

Reviewer #2: No

Reviewer #3: Yes

2. Has the statistical analysis been performed appropriately and rigorously? 

Reviewer #1: Yes

Reviewer #2: I Don't Know

Reviewer #3: Yes

3. Have the authors made all data underlying the findings in their manuscript fully available?

Reviewer #1: Yes

Reviewer #2: Yes

Reviewer #3: Yes

4. Is the manuscript presented in an intelligible fashion and written in standard English?

Reviewer #1: Yes

Reviewer #2: Yes

Reviewer #3: Yes

5. Review Comments to the Author

Reviewer #1: The authors present their findings on an important research topic – attitudes and practices around antimicrobial resistance in Uganda. This is an important study to help inform best practices for antimicrobial stewardship in Africa. However, I think that prior to publishing this work, additional modifications should be made to clarify the methods and results, which should be highlighted further in the discussion. I think the conclusions should also be reframed and toned down.

Study design: Cross-sectional, appropriate to the research question.

Title: Appropriate, though it should be adjusted to note the study is about attitudes in children under 5 if that is the case.

Abstract: Well-written and appropriate.

Introduction: Generally well-written and appropriate to the research material.

Lines 41-44: Can you please specify what information is lacking? Have there ever been surveys of AMS attitudes and practices from sub-Saharan African countries? If so, what were the results? If not, please note this, and also note the other specific gaps in knowledge.

Methods:

Line 53: Consider replacing the word ‘nationwide’ with another word – this study was not conducted at every hospital in Uganda. Specify in the Methods or Results section what proportion of each type of hospital participated in the study.

Lines 58-66: How was the recruitment/invitation to participate carried out? Was any compensation/incentive provided?

Lines 59-62: Why were resident/MMed and other trainee physicians and midwives not included? In some hospitals, I imagine these are the people who may often prescribe or recommend antibiotics to patients.

Lines 69-71: Please describe the sample size method in more detail – referencing the WHO method does not make it clear how the appropriate number of hospitals and number of participants was calculated.

Lines 73-74: Please add the proportion of the total facility type represented by the facilities sampled for this study (e.g. 8/16 RRH = 50%).

Lines 89-90: How was the random sampling performed (e.g. what random selection algorithm was used, and how was it implemented)?

Lines 93-95: Please be more specific about which questionnaire items were adopted from the literature, which have already been validated, and which were created de novo.

Line 110: Why did the authors choose a modified Bloom’s categorization? Has this been validated for this type of research question?

Lines 111-113: I recommend reconsidering the labels of “good” and “bad” for AMS score. These labels imply judgement about attitudes and practices. I recommend changing the labels to “high”, “average”, and “low” or something similar. Labeling people and their attitudes towards AMS as good and bad could have unintended negative consequences.

Line 114: I don’t think gender was assessed, only sex. Is this correct? If so, I recommend changing this to sex.

Lines 133-134: How were discrepancies in doubly-entered data resolved?

Ethics lines 312-313: Why didn’t all research participants give written informed consent to participate? I see a statement at the end of the manuscript that ‘many’ of the participants gave written informed consent, but not all.

I see the Appendix S3 notes the topic of the research is "Antimicrobial stewardship practices and quality of antibacterials use in children under-five in Health facilities of Uganda." Was this study specifically addressing antibacterial use in children? Or all patients?

Results:

Lines 159-160: Were the non-respondents different to the respondents in any way (e.g. professional cadre, sex, age, etc)?

Line 171 and throughout the manuscript: Please see my note about labeling AMS scores as ‘good’ or ‘bad’. The same is true for ‘poor’. I think this should be changed to ‘low’.

Lines 185-187: Please indicate directionality of the association, e.g., which level was positively or negatively associated with higher AMS score.

Line 189: I recommend changing ‘female’ to ‘male’ and reversing the AOR, or changing the description of the association. What is written here implies that being female is associated with ‘good’ AMS attitudes, when in fact the AOR for female was 0.66, so it was ‘protective’ against a ‘good’ AMS attitude (which can be confusing to readers).

Lines 212-213: Similar comment to my prior comment – the authors note that the Western region was a predictor of AMS practices, but the AOR is 0.52, indicating a negative association. Please clarify this in the text to indicate that it was significantly associated with a lower AMS practice score.

Discussion:

Lines 225-231: Are there any other differences between your study and the Ethiopian one that could explain this difference, e.g., years in which the study was carried out, study population, antibiotic availability, etc? Also, this Ethiopian study should probably be mentioned in the introduction as existing data.

Lines 232-233: Please see my note above about indicating the directionality of association. I think it would be clearer to note that male sex was associated with a higher AMS attitude score.

Lines 234-241: The way this association is described is troubling. Firstly, associating sex with good or bad attitudes can be a minefield – see my note above about changing this to high or low scores. Secondly, participant sex is likely confounded by professional cadre, making it difficult to draw inference about this issue. However, the association of participant sex with AMS attitude score was significant while adjusting for level of education, which means that sex is associated with AMS attitude regardless of the level of education. Please examine this interpretation again closely and re-interpret it more carefully.

Additional discussion should be added about why Pharmacists and Pharmacy Techs have high AMS attitude scores but low AMS practice scores – it surprised me to see that their practices were much lower than average and much lower than the other professional cadres.

Additional limitations should be added concerning the lack of data gathered from trainees, and uncertainty around the cause and effect of AMS attitudes and practices.

Conclusion:

The ultimate conclusion – that more needs to be done to support AMS in Uganda – is a good one. However, I don’t think it is justifiable to frame this conclusion in terms of cause and effect – that ‘good’ AMS attitudes are not causing ‘good’ AMS practices – as the authors know, this is association and not causation, and furthermore, there are many systemic barriers to implementing AMS practices regardless of the attitude. This should be acknowledged and the causal language adjusted or removed.

Reviewer #2: This study uses data from interviewer-based surveys administered (from October 2019 to February, 2020) to 582 healthcare providers in 32 (?) facilities to explore attitudes and practices concerning antimicrobial stewardship (AMS) programs in Uganda. The following points will help strengthen the manuscript.

1. Lines 33-50: Could authors add more information to motivate their study of AMS attitudes and practices of healthcare providers? Is this the most important aspect of reducing antibacterial resistance in Uganda? Is the paucity of information on this specific aspect? How was the National Action Plan on Antimicrobial Resistance drafted; bottom-up or top-to-bottom? When drafted? Is it implemented? If yes, how and when (thanks, some information is provided in the Discussion section (lines 227-229) but it needs to be clear from the start) …

2. Lines 53-57: Could authors provide more information about the characteristics of these different hospitals, such as their bed size, teaching status, location characteristics (urban or rural…), their regional distribution etc? Are there any for-profit hospitals in Uganda?

3. Lines 53-57: Could authors explain their reasons for their focus on hospitals only? Are the AMS programs focus only on hospitals?

4. Lines 53-57: Please provide information about the characteristics of regions in Uganda.

5. Lines 69-71: Thanks for the information. This publication requires that at least 3 “sectors” to be identified as a 1st step. So, as before, which are the important “sectors” for the subject matter of this study, only hospitals?

6. Lines 71-74: Thanks for the information. However, it is unclear to the readership how the increase from 4 to 8 regional facilities should be interpreted? The recommendation of “4” is per sector in the publication. Again, are there other “sectors” that are of importance for the subject matter?

7. Lines 74-79: Could authors provide reasons for targeting 24 healthcare providers in each of the hospitals? Are all of these hospitals the same size?

8. Line 82: Again, what are the characteristics of these 4 regions?

9. Lines 82-86: Please be clear if this means that none of the 3 “national referral hospitals” (lines 54-55) were selected into the sample. If so, please provide the rationale.

10. Lines 82-83: Again, how are the 16 regional referral hospitals (line 56) distributed in these 4 regions?

11. Lines 83-85: How are the “50 general (district) hospitals” (line 56) distributed in these 4 regions?

12. Lines 83-85: Please be clear how 3 “general hospitals” in each to the 4 regions add up to “21 general hospitals” (lines 72-73 or line 300) and, therefore, to “32 health facilities” overall (lines 73-74 or Abstract, line 15) for the study?

13. Lines 85-86: Could authors provide information about the regional distribution and characteristics of the 4 “private-not-for-profit health facilities”?

14. Lines 87-88: Please reconcile “departmental heads” here with 4 “heads of departments” above (lines 77-78).

15. Lines 88-90: Again, do all the facilities have the same number of healthcare providers?

16. Lines 130-135: Please be clear if this means that there were no missing data items. If there were missing data items, what procedures were followed?

17. Lines 159-160: Again, please address “768” potential respondents in light of the number of hospitals included in the study sample. As above, what was the number of hospitals in the sample; “32” or 23?

18. Lines 159-218: The reviewer finds it impossible to interpret the results without clarifications for the points above.

19. Lines 227-229: Again, could authors make clear from the very beginning what was the rationale for their study, in general, and the survey, especially the “AMS practice” part, if “the Ministry of Health in Uganda has not implemented any formal interventions on AMS among healthcare providers”?

20. Please avoid typos and ensure completeness and transparency in the manuscript: a) please ensure that all of the acronyms (such as, AMS, in line 25) are spelled out the first time they are mentioned (it is in the Abstract but also needs to be spelled out in the body of the manuscript), b) please ensure that the references are complete (such as #18, for example, please be clear about “WHO” and the location of the publisher…), c) “would; reduce”(?) in line 173 or “is has been”(?) in line 257, d) please ensure that tables are self-explanatory (see, for example, Table 2, what are “AMS” and “60 points” & what does the second note refer to, or Table 4, why “HCP”…)

Reviewer #3: Thank you very much for the opportunity to review this research.

The study investigated healthcare providers' attitude and practices on AMS and associated factors in regional referral hospitals, general hospitals, and private-not-for-profit health facilities in Uganda.

The manuscript is technically sound, and data support the conclusions.

It requires minor corrections. Here are some minor suggestions for improvement.

INTRODUCTION

Please do not start the sentence with an acronym. Also, specify what AMS stands for.

I would suggest organizing the introduction better. I think that it is a little confusing, too many acronyms, and the concepts are mixed. It would be appropriate to outline the context the research refer to and then mention what happened in other countries. Subsequently, I would analyze the problem spread of the antibacterial resistance and the importance of AMS programmes in Uganda.

METHODS

Why did the authors exclude who had worked for only one year in the health facility? What was the rationale?

Authors used ordinal logistic regression to model the data (outcomes were divided into 3 categories). Was the proportional odds assumption checked?

How were variables selected into the models? How was model fit assessed?

DISCUSSION

In the discussion section it seems redundant to define what was done in this study and repeat the results. It is correct instead the comparison with other findings, as the authors did, even though this section should be expanded to present a broader context in which this research is relevant (there are only a few citations to Ethiopia). For instance, results should be interpreted within the perspective of antimicrobial resistance and healthcare-associated infections, considering both low-middle income countries and around the world. Relevant articles worth including may be PMIDs 34223045, 32062724, 33961678, 33882843, 29590400, and 34213520.

6. PLOS authors have the option to publish the peer review history of their article (what does this mean?). If published, this will include your full peer review and any attached files.

Reviewer #1: No

Reviewer #2: No

Reviewer #3: No

---

## [Author Response · Author response to Decision Letter 0]

6 Sep 2021

We thank you for offering us an opportunity to share our results with the entire research community.

---

## [Decision Letter · Decision Letter 1]

5 Oct 2021

PONE-D-21-10939R1Antimicrobial stewardship: Attitudes and practices of healthcare providers in selected health facilities in UgandaPLOS ONE

Dear Dr. Kimbowa,

Thank you for submitting your manuscript to PLOS ONE. After careful consideration, we feel that it has merit but does not fully meet PLOS ONE’s publication criteria as it currently stands. Therefore, we invite you to submit a revised version of the manuscript that addresses the points raised during the review process.

Thank you to the authors for their work on revising the manuscript so far. Could the authors now focus on the point below, explaining what choices they made when administering the surveys and clarifying the rationale behind such choices?

The manuscripts provides a good amount of information about, for example, the healthcare worker population, hospitals, and regions. It does not appear as though this info has been used to devise an appropriate cluster sampling, which would have in turn guided the analysis of data.

Although the authors calculated an overall sample size, this was not used to administer the surveys randomly around the country, as the administration ultimately does not appear random, but rather distributed to certain (types of) hospitals in some regions.

We look forward to receiving your revised manuscript.

Kind regards,

Elena Ambrosino

Academic Editor

PLOS ONE

Reviewers' comments:

Reviewer's Responses to Questions

**Comments to the Author**

1. If the authors have adequately addressed your comments raised in a previous round of review and you feel that this manuscript is now acceptable for publication, you may indicate that here to bypass the “Comments to the Author” section, enter your conflict of interest statement in the “Confidential to Editor” section, and submit your "Accept" recommendation.

Reviewer #1: (No Response)

Reviewer #2: (No Response)

2. Is the manuscript technically sound, and do the data support the conclusions?

Reviewer #1: Yes

Reviewer #2: No

3. Has the statistical analysis been performed appropriately and rigorously? 

Reviewer #1: Yes

Reviewer #2: I Don't Know

4. Have the authors made all data underlying the findings in their manuscript fully available?

Reviewer #1: Yes

Reviewer #2: Yes

5. Is the manuscript presented in an intelligible fashion and written in standard English?

Reviewer #1: Yes

Reviewer #2: Yes

6. Review Comments to the Author

Reviewer #1: This revised version is significantly improved from prior. I recommend some additional minor changes prior to publication, which I have noted below.

Title and Abstract: Unchanged from prior, no changes recommended.

Introduction: The authors have largely re-written the introduction, and it is well-written. No further changes recommended.

Methods and Results:

Largely re-written. There is excessive detail on the number of hospitals in each region which is somewhat distracting to the reader. The authors still have not described how recruitment/invitation to participate was carried out at each facility to reach the enrollment targets, and whether any compensation/incentive was provided.

In my prior comments, I suggested categorizing the AMS attitude scores as ‘high’, ‘average’, and ‘low’. Re-reading the manuscript now, I think ‘average’ is not the right label, as this has statistical meaning as well. Consider using ‘moderate’ or ‘fair’ to label this category instead of ‘average’.

Results of statistical models have been re-written and are more appropriate. In Line 348, I recommend changing the start of the sentence to ‘In multivariable models adjusting for possible confounders, AMS practice scores …’ This is to clarify that you are referring to your model results. Also include the AOR and its associated confidence interval in this sentence, not just the P-value for the association.

Discussion and Conclusions:

Largely re-written. In line 364 I recommend avoiding the phrase ‘will never’, because this is unlikely to be the case. Consider rephrasing as ‘are unlikely to’. In addition, there is only one limitation addressed – about potential bias in the interview. I think additional limitations should be added concerning the lack of data gathered from trainees (though part-time employees, they do prescribe a significant proportion of antibiotics and have their own AMS attitudes and practices, which were not measured here), uncertainty around the cause and effect of AMS

attitudes and practices, and potential lack of generalizability of your results to non-Ugandan or non-East African settings.

Reviewer #2: Thanks for an improved manuscript. The following needs to be taken into consideration.

1. “Old Lines 33-50: Could authors add more information to motivate their study of AMS attitudes and practices of healthcare providers? Is this the most important aspect of reducing antibacterial resistance in Uganda? Is the paucity of information on this specific aspect? How was the National Action Plan on Antimicrobial Resistance drafted; bottom-up or top-to-bottom? When drafted? Is it implemented? If yes, how and when (thanks, some information is provided in the Discussion section (lines 227-229) but it needs to be clear from the start) …”

Thanks for revisions; the Introduction section motivates the study well. However, it is now more than 3 pages long. Please consider shortening it without losing any of its content.

2. “Old Lines 53-57: Could authors provide more information about the characteristics of these different hospitals, such as their bed size, teaching status, location characteristics (urban or rural…), their regional distribution etc? Are there any for-profit hospitals in Uganda?”

Thanks for providing more information. This section needs to be shortened and, please, just provide information that is relevant to motivate the sampling strategy. Specific questions: (a) Thanks for information about 2 (or is it 3?) “national referral” hospitals (new lines 102-105) that were excluded. As a result, (and names are not required but) what % of the public health hospitals were excluded? (b) “two million” (line 107)? (c) Which region is this; (again, we do not need names and the detail but) why “five regional referral hospitals” followed with 4 names (lines 108-109)? (d) What does “approximately 50 general hospitals” (line 116) mean or are there 41 (lines 115-124) general hospitals? (e) Please be clear, do these hospitals have about 10 beds each (lines 123-124) or 100 beds? (f) If there are “44” private not-for-profit (PNFP) hospitals and if “3 out of 4” that are considered as “regional referral centres” were included in this study, what % of PNFPs was excluded from this study?

3. “Old Lines 53-57: Could authors explain their reasons for their focus on hospitals only? Are the AMS programmes focus at hospitals?”

Thanks for explanations. In the healthcare system, are hospitals the only healthcare institution type where antibacterial misuse is observed?

5. “Old Lines 69-71: Thanks for the information. This publication requires that at least 3 “sectors” to be identified as a 1st step. So, as before, which are the important “sectors” for the subject matter of this study, only hospitals?”

This section in its new format now indicates that information about total number of observational units were available to authors. Was this information also available at the hospital level? If yes, why the sampling was not conducted at the cluster level? Thanks for indicating later that the sample consists mostly of same size “general hospitals” but what about the sizes of other hospitals in the sample (for example is the 1000 bed “regional referral” hospital in the sample)?

6. “Old Lines 71-74: Thanks for the information. However, it is unclear to the readership how the increase from 4 to 8 regional facilities should be interpreted? The recommendation of “4” is per sector in the publication. Again, are there other “sectors” that are of importance for the subject matter?”

Thanks for revisions. If “4 general hospitals for each” of the 8 “regional referral facility” was selected, how did the authors end up with “21 general hospitals” and with the regional distribution of the sample provided (lines 162-165)? Also, see above for what “approximately 50 general hospitals” (line 116) mean.

9. “Old Lines 82-86: Please be clear if this means that none of the 3 “national referral hospitals” (lines 54-55) were selected into the sample. If so, please provide the rationale.”

Thanks for information about 2 “national referral” hospitals (new lines 102-105) that were excluded. Is there a 3rd one?

12. “Old Lines 83-85: Please be clear how 3 “general hospitals” in each to the 4 regions add up to “21 general hospitals” (lines 72-73 or line 300) and, therefore, to “32 health facilities” overall (lines 73-74 or Abstract, line 15) for the study?”

Please see above; “21 general hospitals” is still not clear to the this reviewer.

18. “Old Lines 159-218: The reviewer finds it impossible to interpret the results without clarifications for the points above.”

This reviewer still finds it impossible. Did authors analyze their data as if it was a random sample?

20. Old “Please avoid typos and ensure completeness and transparency in the manuscript: a) please ensure that all of the acronyms (such as, AMS, in line 25), are spelled out the first time they are mentioned (it is in the Abstract but also needs to be spelled out in the body of the manuscript), b) please ensure that the references are complete (such as #18, for example, please be clear about “WHO” and the location of the publisher…), c) “would; reduce”(?) in line 173 or “is has been”(?) in line 257, d) please ensure that tables are self-explanatory (see, for example, Table 2, what are “AMS”, “60 points” and what does the second note refer to, or Table 4, why “HCP”…)”

Some of the new points: a) new line 27, do you mean “arises”, b) new line 33, the sentence that starts with “Antimicrobial stewardship (AMS)…” needs to be a new paragraph, c) new line 61, “where” seems to be missing, d) “LMIC”? new line 76….

7. PLOS authors have the option to publish the peer review history of their article (what does this mean?). If published, this will include your full peer review and any attached files.

Reviewer #1: No

Reviewer #2: No

---

## [Author Response · Author response to Decision Letter 1]

15 Nov 2021

RESPONSE TO REVIEWER'S COMMENTS ON MANUSCRIPT, PONE-D-21-10939R1

Dear Editor and reviewers: 

We greatly appreciate the thorough and thoughtful comments provided on our submitted manuscript. We have revised our manuscript according to the reviewers' comments, questions, and suggestions. Please see below our detailed responses to the reviewers' comments. All our responses are in bold. Please let us know if you still have any questions or concerns about the manuscript, and we will be happy to address them promptly. 

Yours, 

The authors of PONE-D-21-10939R1

Editor's comment:

Thank you to the authors for their work on revising the manuscript so far. Could the authors now focus on the point below, explaining what choices they made when administering the surveys and clarifying the rationale behind such choices?

Response to Editor's comment #1:

We choose different types of health facilities for this study because of a gap in the literature on attitudes and practices of healthcare providers between urban regional referral hospitals, private not for profit (PNFP) hospitals and community general hospitals. Previous studies in Nigeria and Ethiopia had concentrated on studying only tertiary or specialised hospitals, limiting their generalizability. The study hospitals were selected due to the presence of structures and systems for strengthening the monitoring and improvement of medicine use. Most government and development partners capacity building has focused on establishing infection prevention committees, medicines and therapeutic committees in these hospitals. 

We chose a large sample size of healthcare providers to increase the study's representativeness and determine the predictors of AMS attitude and practices in health facilities in Uganda. 

We chose the administration office and heads of department to recruit the healthcare providers into our study. We used emails, letters, and phone calls to recruit healthcare providers to increase the participation of healthcare providers. 

Every participant consented first before being interviewed.

We used an interview-administered questionnaire. We used research assistants from the same facility to administer the questionnaire to minimise bias in the responses. During piloting, we realised respondents gave different responses when the principal investigator or research assistant interviewed, so we used a research assistant other than the investigator himself to minimise the measurement bias. We used research assistants from different health backgrounds within the health facility to avoid the interviewer social desirability effect. 

Editor's comment:

The manuscripts provides a good amount of information about, for example, the healthcare worker population, hospitals, and regions. It does not appear as though this info has been used to devise an appropriate cluster sampling, which would have in turn guided the analysis of data.

Response to Editor's comment #2: 

However, much as we couldn't conduct cluster sampling, the study catered for clustering at the healthcare provider level at the health facility 

Editor's comment:

Although the authors calculated an overall sample size, this was not used to administer the surveys randomly around the country, as the administration ultimately does not appear random, but rather distributed to certain (types of) hospitals in some regions.

Response to Editor's comment #3:

Yes, we agree with the Editor's comments. We could not use random sampling to all types of hospitals since we had to receive administrative clearance to know the sampling frame of participating hospitals. We randomly selected eight out of 16 regional referrals, where two regional referrals were picked per region to increase regional representativeness. All eight out 16 had granted us administrative clearance. However, of all the 32 out of 47 general hospitals selected after random sampling, only 21 granted us administrative clearance to conduct the study. Eleven general hospitals didn't participate in the selected sample.

We could not randomly select pharmacists, pharmacy technicians, and laboratory technicians in all hospitals that participated since each hospital had only one employee of these types of professional cadre. However, nurses, medical officers, clinical officers and medical specialists were selected randomly. 

Reviewer #1

This revised version is significantly improved from prior. I recommend some additional minor changes prior to publication, which I have noted below.

Response: 

Thanks for the valuable comments, and we appreciate your time. 

Methods and Results:

Largely re-written. There is excessive detail on the number of hospitals in each region which is somewhat distracting to the reader. 

Response: 

Lines 99-107 on page 6; We have improved the clarity of writing in this section, deleted names of hospitals and limited ourselves to the number of targeted hospitals and the reasons for their selection. 

The authors still have not described how recruitment/invitation to participate was carried out at each facility to reach the enrollment targets and whether compensation/incentive was provided.

Response:

Lines 197-205, page 11. We apologise for that, and we have added more information on how the research assistant conducted recruitment. We have also added more information on compensation for time and transport since some respondents were on leave and were requested to come to the facility

In my prior comments, I suggested categorising the AMS attitude scores as 'high', 'average', and 'low'. Re-reading the manuscript now, I think 'average' is not the right label, as this has statistical meaning as well. Consider using 'moderate' or 'fair' to label this category instead of 'average'.

Response: 

The authors want to thank the review and have agreed to use "fair" instead of "average", as reflected in this manuscript's sections. 

Comment Reviewer #1: 

Results of statistical models have been re-written and are more appropriate. In Line 348, I recommend changing the start of the sentence to 'In multivariable models adjusting for possible confounders, AMS practice scores …' This is to clarify that you are referring to your model results. Also include the AOR and its associated confidence interval in this sentence, not just the P-value for the association.

Response: 

Lines 321-323, page 23-24. We have corrected it as suggested. 

Comment Reviewer #1: 

Discussion and Conclusions:

Largely re-written. In line 364 I recommend avoiding the phrase 'will never', because this is unlikely to be the case.

Response: 

Lines 333-334, pages 25, Thanks for the correction. We have improved the clarity of the statement 

Consider rephrasing as 'are unlikely to'. In addition, there is only one limitation addressed – about potential bias in the interview. I think additional limitations should be added concerning the lack of data gathered from trainees (though part-time employees, they do prescribe a significant proportion of antibiotics and have their own AMS attitudes and practices, which were not measured here), uncertainty around the cause and effect of AMS attitudes and practices, and potential lack of generalizability of your results to non-Ugandan or non-East African settings.

Response: 

Lines 417-419, page 29. We have included the limitation of part and intern healthcare providers. 

Lines 422-423, Pages 29. We have added a limitation on the uncertainty surrounding the cause and effect of AMS attitudes and practices.

Lines 423-424, Pages 29: We have added an aspect on our results not being generalisable to other non-East African settings or non-Ugandan setting 

Reviewer #2

Reviewer #2: Thanks for an improved manuscript. The following needs to be taken into consideration.

Response: 

Thanks for the valuable comments; we appreciate

Comment Reviewer #2: 

Thanks for revisions; the Introduction section motivates the study well. However, it is now more than three pages long. Please consider shortening it without losing any of its content.

Response: 

Lines 26-92, We have shortened the introduction section and its now two pages. 

Comment Reviewer #2: 

Thanks for information about 2 (or is it 3?) "national referral" hospitals (new lines 102-105) that were excluded. a)As a result, (and names are not required but) what % of the public health hospitals were excluded?

Response: 

Lines 86-107, pages 6. We have clarified the health system characteristics. In our clarification, we report that the health system comprises 6937 health facilities, and of these, 3133 (45.6%) are public health facilities. The composition of the latter includes two national referral hospitals, three referral hospitals (the Uganda Cancer Institute, the Uganda Heart Institute, and the Women's Hospital), 13 regional referral hospitals, 47 general hospitals, 166 level IV health centres, 962 level III health centres, and 1321 level II health centres. We had only 63 out of 3133 (2%) health facilities participating. We excluded 3070 health facilities. The percentage of health facilities excluded is 98% (3070/3133). 

(a) "two million" (line 107)?

Response: 

Lines 86-107, pages 6. Thanks for this highlight. According to the health facilities of Uganda Master list (2018) page 7, Each health facility has a designated population size that it is meant to serve; a national referral hospital (10,000,000 persons), each regional referral hospital (2,000,000 persons), general hospitals (500,000) and, Health centre IV (100,000), Health centre III (20000), health centre II (5000) and health centre I (not defined).

In our previous description of the study setting, we had included this characteristic showing that a regional referral serves over two million persons; however, we have removed it in the revised manuscript for clarity. 

 (c) Which region is this; (again, we do not need names and the detail but) why "five regional referral hospitals" followed with 4 names (lines 108-109)? 

Response: 

Lines 99-107, pages 6. We thank you for this observation and the need for clarification. In the central region, we have five regional referrals hospitals that we included in our study. 

Comment Reviewer #2: 

(d) What does "approximately 50 general hospitals" (line 116) mean or are there 41 (lines 115-124) general hospitals? (e) Please be clear, do these hospitals have about 10 beds each (lines 123-124) or 100 beds? (f) If there are "44" private not-for-profit (PNFP) hospitals and if "3 out of 4" that are considered as "regional referral centres" were included in this study, what % of PNFPs was excluded from this study? 

Response: 

Lines 95-107, page 6. In the PNFP health systems, there are 1009 health facilities with the highest referral level in the four tertiary hospitals, followed by 40 general hospitals and 955 health centres. We excluded all the 40 general PNFP hospitals and 955 health centres making a percentage of 98.6%.

Comment Reviewer #2: 

3. "Old Lines 53-57: Could authors explain their reasons for their focus on hospitals only? Are the AMS programmes focus at hospitals?"

Thanks for explanations. In the healthcare system, are hospitals the only healthcare institution type where antibacterial misuse is observed?

Response: 

Lines 102-108, page 6. Thanks for the comment. We agree with you that overuse and misuse of antibacterials are not limited to regional referrals, general hospitals and PNFPs. Recent health initiatives have focused on training most public health facilities, tertiary PNFP healthcare providers, and private healthcare providers in communities on antimicrobial resistance and antimicrobial stewardship. Indeed most healthcare providers have been trained on the implementation of antimicrobial stewardship interventions. There has been the ongoing implementation of the National action plan, and the focus of the government has been on strengthening existing or initiating medicines and therapeutic committees. The committees implement antimicrobial stewardship, pharmacovigilance, and supply chain management in regional referrals, general hospitals, and PNFPs. Our study focused on assessing the attitudes and practices of healthcare providers on the implementation of these programs. However, further initiatives are needed to understand the attitudes and practices of healthcare workers in the lower health facilities since there are many challenges like infrastructure, human resources and follow of information. 

Comment Reviewer #2: 

This section in its new format now indicates that information about total number of observational units were available to authors. Was this information also available at the hospital level? 

Response: 

Lines 130-141, page 8. Thanks for this comment. We could not know how many hospitals we shall collect from data not until the hospitals granted ethical and administrative clearance. We did not have information at the hospital level because the ethical guidelines in Uganda required the researcher to be issued ethical clearance from the parent institution and national ethical clearance from the national institution. Finally, administrative clearance to all respective hospitals we targeted before being availed of any information at the hospital level. No respondent can participate in the study without any administrative clearance. After selecting eight out of 16 regional referrals, clearance was granted by all the selected eight regional referrals hospitals. After selecting 32 out of the 47 general hospitals, administrative clearance was granted by only 21 hospitals. 

If yes, why the sampling was not conducted at the cluster level? 

Response: 

Lines 130-141, page 8. The sampling procedure did not take a cluster sampling approach because of a shortfall in the participation of hospitals in the northern region of the country (seven general hospitals did not grant us administrative clearance). The cluster of North would have had one general hospital compared to six general hospitals in the Central, seven general hospitals in Eastern and seven in the western. There was an imbalance in terms of general hospitals. The North would not form a cluster because of the non-participation of six hospitals. Therefore, there would be a shortfall in the study population (healthcare providers) representation northern region. Furthermore, this would mean there would be one cluster with no similar characteristics in terms of healthcare providers or participating types of hospitals compared to other regions. 

However, since there was clustering among healthcare providers in terms of cadres at the lowest level, the study adjusted for clustering in the design effect of sample size determination also adjusted for clustering during analysis using robust standard errors. 

Comment Reviewer #2: 

Thanks for indicating later that the sample consists mostly of same size "general hospitals" but what about the sizes of other hospitals in the sample (for example is the 1000 bed "regional referral" hospital in the sample)? 

Response: 

Lines 143-149, page 8. Thanks for this comment. The study ensured proportionate representation among participating hospitals by using a proportionate number to size to select all other professional cadres except pharmacists, pharmacy technicians and laboratory technicians because each hospital had one cadre. The latter were purposively selected in each participating hospital.

Comment Reviewer #2: If "4 general hospitals for each" of the 8 "regional referral facility" was selected, how did the authors end up with "21 general hospitals" and with the regional distribution of the sample provided (lines 162-165)? Also, see above for what "approximately 50 general hospitals" (line 116) mean.

 If "4 general hospitals for each" of the 8 "regional referral facility" was selected, how did the authors end up with "21 general hospitals" 

Response: 

Lines 133-141, page 8. Thanks for this question. Before data collection, we selected randomly eight regional referrals hospitals. We selected 32 general hospitals out of the 47 included general hospitals. For each of the selected regional referrals, we had to select four general hospitals. Therefore for all the four regions (North, West, East and Central), we selected 32 general hospitals out of the 47. How did we end up with 21 general hospitals? After selecting the eight regional referrals, 32 general hospitals, and three tertiary PNFPs, we applied for administrative clearance in each selected health facility. All the eight regional referrals and three tertiary PNFPs granted us administrative clearance. However, for general hospitals, only 21 out of 32 granted us administrative clearance?. The ethical obligation by the Uganda National Council of Science and Technology and the parent institution (Makerere University Higher degrees ethics and research committees and the hospital research and ethics committee) can only be done once granted ethical clearance by all the three institutions. That is how 21 general hospitals appear in the stated methods.

Also, see above for what "approximately 50 general hospitals" (line 116) mean.

Lines 91-92, page 6. Thanks for this question, and we apologies for not being precise on using "approximately 50" general hospitals. According to the national census for health facilities carried out in 2014 and the national health facility master list, there are 47 general hospitals in Uganda. 

Comment Reviewer #2: 

Thanks for information about 2 "national referral" hospitals (new lines 102-105) that were excluded. Is there a 3rd one?

Response: 

Lines 91-92, page 6. By 2014 when a national census for health facilities was done, there were three national referrals, Mulago super specialised hospitals, Butabika national referral for mental health, and Buluba national referral for leprosy. In the latest health facility master list 2018, the Ministry of Health no longer recognises Buluba as a national referral, and it implies we are left with two national referral hospitals. 

Comment Reviewer #2: 

Please see above; "21 general hospitals" is still not clear to the this reviewer.

Response: 

Lines 144-156, page 8. Thanks for this question. Before data collection, we selected randomly eight regional referrals. Then we selected 32 of the 47 general hospitals, where for each of the selected regional referrals, we had to select four general hospitals. We selected 32 general hospitals out of the 47. How did we end up with 21 general hospitals? After selecting the 32 general hospitals, we applied for administrative clearance in each of the selected general hospitals. Only 21 general hospitals granted us the clearance to conduct the study with them. These included one in the North, six in the central, seven in the East and seven in the West. The ethical obligation to the Uganda National Council of Science and Technology and the Makerere University Higher degrees ethics and research committee, and the hospital research and ethics committee was that the researcher could only conduct data collection after all three institutions had received ethical clearance. That is how the 21 general hospitals appear stated in our methods.

18. "Old Lines 159-218: The reviewer finds it impossible to interpret the results without clarifications for the points above."

Response :

Lines 132-141, page 8. After clarifying the information on the number of hospitals that participated and how they were selected we believe the reviewer will agree with us on this revision that results can be easily interpreted

Comment Reviewer #2: 

This reviewer still finds it impossible. Did authors analyse their data as if it was a random sample?

Response: 

The answer is a yes or no. We had chosen facilities randomly, but the healthcare providers like pharmacists, laboratory technicians and pharmacy technicians were chosen purposively since we only found one person per health facility. In contrast, nurses, medical officers, medical specialists, and clinical officers all were chosen by simple random sampling using sampling frames of the hospital facilities. Having a sample with a mixture of persons who were picked purposively and others randomly the answer may lie most on the no side.

Comment Reviewer #2: 

20. Old "Please avoid typos and ensure completeness and transparency in the manuscript:

Response 

Thanks for this observation, we have corrected all typos in the manuscript

a) please ensure that all of the acronyms (such as, AMS, in line 25), are spelled out the first time they are mentioned (it is in the Abstract but also needs to be spelled out in the body of the manuscript),

Response 

Lines 2, page 1 and Lines 36, page 2. Thanks for the correction, We have spelt out the word AMS twice in the abstract and once in the main section

b) please ensure that the references are complete (such as #18, for example, please be clear about "WHO" and the location of the publisher…), 

Response 

Lines 498-601: Thanks for the correction, and we have ensured the references are all complete.

c) "would; reduce" (?) in line 173 or "is has been" (?) in line 257, d) please ensure that tables are self-explanatory (see, for example, Table 2, what are "AMS", "60 points" and what does the second note refer to, or Table 4, why "HCP"…)"

Response 

Lines 271 pages 16. Thanks for the correction, "would reduce" corrected to "minimises."

Lines 390 pages 27. "is has been" corrected to "has been".

Table 2: We reported the mean score in 12 questions on antimicrobial stewardship attitudes. This has been improved.

Table 4: Sorry for the HCP. It has been corrected. 

Some of the new points: 

a) new line 27, do you mean "arises", 

Response 

Lines 28-29: we corrected the statement.

b) new line 33, the sentence that starts with "Antimicrobial stewardship (AMS)…" needs to be a new paragraph,

Response

Lines 37: We thank you for this suggestion. We have made this correction.

b) new line 61, "where" seems to be missing, d) "LMIC"? new line 76….

Lines 73, pages 2: we have corrected the abbreviation 

7. PLOS authors have the option to publish the peer review history of their article (what does this mean?). If published, this will include your full peer review and any attached files.

No

Do you want your identity to be public for this peer review? For information about this choice, including consent withdrawal, please see our Privacy Policy.

Reviewer #1: No

Reviewer #2: No

 No

---

## [Editor Report · Decision Letter 2]

23 Nov 2021

PONE-D-21-10939R2Antimicrobial stewardship: Attitudes and practices of healthcare providers in selected health facilities in UgandaPLOS ONE

Dear Dr. Kimbowa,

Thank you for submitting your manuscript to PLOS ONE. After careful consideration, we feel that it has merit but does not fully meet PLOS ONE’s publication criteria as it currently stands. Therefore, we invite you to submit a revised version of the manuscript that addresses the points raised during the review process.

Please carefully consider the detailed feedback from the reviewers. The revision is an improvement by several methodological details have not been clarified yet. The authors have still a chance to work on these improvements.

We look forward to receiving your revised manuscript.

Kind regards,

Elena Ambrosino

Academic Editor

PLOS ONE
---

## [Author Response · Author response to Decision Letter 2]

6 Jan 2022

I thank you for the time and comments on this manuscript, they have improved the manuscript.

---

## [Editor Report · Decision Letter 3]

11 Jan 2022

Antimicrobial stewardship: Attitudes and practices of healthcare providers in selected health facilities in Uganda

PONE-D-21-10939R3

Dear Dr. Kimbowa,

We’re pleased to inform you that your manuscript has been judged scientifically suitable for publication and will be formally accepted for publication once it meets all outstanding technical requirements.

Kind regards,

Elena Ambrosino

Academic Editor

PLOS ONE
---

## [Editor Report · Acceptance letter]

25 Jan 2022

PONE-D-21-10939R3 

Antimicrobial stewardship: Attitudes and practices of healthcare providers in selected health facilities in Uganda 

Dear Dr. Kimbowa:

I'm pleased to inform you that your manuscript has been deemed suitable for publication in PLOS ONE. Congratulations! Your manuscript is now with our production department. 

Kind regards, 

on behalf of

Dr. Elena Ambrosino 

Academic Editor

PLOS ONE